# QTL Analysis of β-Glucan Content and Other Grain Traits in a Recombinant Population of Spring Barley

**DOI:** 10.3390/ijms25126296

**Published:** 2024-06-07

**Authors:** Alberto Gianinetti, Roberta Ghizzoni, Francesca Desiderio, Caterina Morcia, Valeria Terzi, Marina Baronchelli

**Affiliations:** Council for Agricultural Research and Economics (CREA), Research Centre for Genomics and Bioinformatics, Via S. Protaso 302, 29017 Fiorenzuola d’Arda, PC, Italy; roberta.ghizzoni@crea.gov.it (R.G.); francesca.desiderio@crea.gov.it (F.D.); caterina.morcia@crea.gov.it (C.M.); valeria.terzi@crea.gov.it (V.T.); marina.baronchelli@crea.gov.it (M.B.)

**Keywords:** β-glucan, transgressive segregation, *HvCslF9*

## Abstract

Barley with high grain β-glucan content is valuable for functional foods. The identification of loci for high β-glucan content is, thus, of great importance for barley breeding. Segregation mapping for the content in β-glucan and other barley grain components (starch, protein, lipid, ash, phosphorous, calcium, sodium) was performed using the progeny of the cross between Glacier AC38, a mutant with high amylose, and CDC Fibar, a high β-glucan waxy cultivar. The offspring of this cross showed transgressive segregation for β-glucan content. Linkage analysis based on single-nucleotide polymorphism (SNP) molecular markers was used for the genotyping of the parents and recombinant inbred lines (RILs). Two Quantitative Trait Loci (QTL) for β-glucan content and several QTL for other grain components were found. The former ones, located on chromosomes 1H and 7H, explained 27.9% and 27.4% of the phenotypic variance, respectively. Glacier AC38 provided the allele for high β-glucan content at the QTL on chromosome 1H, whereas CDC Fibar contributed the allele at the QTL on chromosome 7H. Their recombination resulted in a novel haplotype with higher β-glucan content, up to 18.4%. Candidate genes are proposed for these two QTL: *HvCslF9*, involved in β-glucan biosynthesis, for the QTL on chromosome 1H; *Horvu_PLANET_7H01G069300*, a gene encoding an ATP-Binding Cassette (ABC) transporter, for the QTL on chromosome 7H.

## 1. Introduction

Although barley (*Hordeum vulgare* L.) is mainly used as feed, its kernels are also consumed as human food, especially as an ingredient in soups [1]. The barley grain has some potential for use by the food industry because of its high content of β-glucan, a kind of dietary fiber that makes it suitable to produce foods with healthy functions, i.e., functional foods [2,3,4,5].

The β-glucan, chemically a population of (1→3)-(1→4) mixed-linked β-D-glucans, is a non-starchy polysaccharide representing the main structural component of the endosperm cell walls in cereals like barley and oat [6,7,8]. Differently from non-cereal β-glucans, cereal β-glucan is unsubstituted and unbranched [4,5,9,10]. Barley β-glucan is a high-molecular-weight polymer of β-D-glucopyranose whose linkages are about 30% β-(1→3) and 70% β-(1→4) [8]. The (1→4)-linked β-D-glucopyranosyl residues (the cellulose-like portions) of the polymeric molecule occur in groups, mostly of three or four, separated by single (1→3) linkages [8,10]. This irregular structure makes β-glucan much more flexible than cellulose, and soluble in the gut under physiological conditions [5], even though barley β-glucan is only partially soluble in pure water, wherein, due to its high water-binding capacity, it forms highly viscous solutions [5,8]. In general, uniform glucan chains have a tendency for intermolecular aggregation—and consequently they are poorly soluble—because the regularity of these chains favors the formation of many hydrogen bonds among identical chains. In β-glucan, β-(1→3) glycosidic bonds break the regularity of the sequences of β-(1→4) linkages and, thus, reduce the formation of hydrogen bonds and the aggregation among chains, thereby increasing solubility in water [8]. As consumed food items contain several kinds of polysaccharides, the simultaneous presence in the gut of diverse polymeric carbohydrates allows the formation of heterogeneous interactions. This further reduces the probability of intermolecular aggregation between β-glucan chains, which are already only weakly aggregated, thereby facilitating their thorough solubilization [5]. Because of their capability to establish many hydrogen bonds, however, β-glucan chains, when ingested at a sufficient concentration, form a net that reduces enzyme accessibility and, therefore, digestion of starch [5,11]. This net can also trap glucose and cholesterol, decreasing their absorption in the gut [3]. Barley β-glucan is, therefore, soluble dietary fiber [5,12], which, in adequate amounts, is effective in lowering blood cholesterol, glycemic index and preventing cardiovascular disease [2,3,13,14]. The inclusion of β-glucan in a wide range of both cereal and dairy-based foods has gained interest [1,2,3] because, in developed countries, people tend to consume excessive amounts of carbohydrates, protein and lipid, which increase their risk of suffering from chronic diseases like obesity, type 2 diabetes mellitus and coronary heart disease [15].

Barley cultivars commonly contain from about 4% to 10% grain β-glucan, though some low-starch mutants (with shrunken caryopses) reach contents as high as 20% [8]. Genotypes with very low β-glucan content, <1%, have been described too [16]. The level of β-glucan in the grain is strongly dependent upon the genotype [2,8] whereas the effects of the environment and the interaction between environments and genotypes are typically smaller [17]. It has nevertheless been reported that up to about half the variability in β-glucan content can be due to environmental effects and their interactions with genotype [18].

Large genetic increases in β-glucan content chiefly depend on restrictions in starch biosynthesis that lead to the diversion of carbon into β-glucan synthesis rather than on the physiological regulation of β-glucan biosynthesis [8]. In the Poaceae, two *Cellulose synthase-like* (*Csl*) gene subfamilies, *CslF* and *CslH*,—which belong to the *Cellulose synthase* superfamily that is responsible for the synthesis of several plant cell wall polysaccharides—are responsible for the synthesis of (1→3)-(1→4) mixed-linked β-D-glucan, with a major role for the *CslF* subfamily [8]. In barley, the *HvCslF* gene subfamily is comprised of ten members [19].

In the starchy endosperm of developing barley grains, *HvCslF6* and *HvCslF9* are the predominant *Csl* transcripts [20]. Accordingly, expression analysis of *HvCslF* and *HvCslH* gene family members identified *HvCslF6* and *HvCslF9* as the major β-glucan synthase genes expressed in various stages of barley grain development [21]. *HvCslF6* is expressed in almost all tissues and during the entire grain development, though it reaches maximum expression in the developing endosperm later than two weeks after pollination, whereas *HvCslF9* is expressed in the coleoptile, roots, and developing seed 4–12 days after pollination [20]. Based on the expression studies [20,21,22] and a study using a reverse genetics approach employing CRISPR/Cas9 to generate mutations in members of the *Cellulose synthase-like* (*Csl*) gene family [23], *HvCslF6* was concluded to be the major *Cellulose synthase-like* gene determining β-glucan content in barley grains, while *HvCslF9* appeared to have a minor role. The role of CslF6 as the key enzyme controlling the biosynthesis of β-glucan in barley was demonstrated by the fact that, when chemical mutagenesis was used to induce an inactivating lesion in the *HvCslF6* gene, a β-glucan-less barley line was obtained [24].

Although *Csl* genes are responsible for β-glucan synthesis, many of the very high β-glucan-containing barleys have mutations in starch biosynthetic genes, suggesting that, in low-starch mutants, defective starch synthesis causes grain glucose to be redirected to β-glucan synthesis [8,25,26,27]. This was also confirmed with a reverse approach showing that over-expressing the *HvCslF6* gene under the control of an endosperm-specific promoter caused β-glucan content to increase almost two-fold in the transgenic grain, while starch decreased dramatically [28]. Starch and β-glucan syntheses in the cereal grain are, therefore, complementary, which causes an inverse relationship between starch and β-glucan content in the mature barley grain [8].

Correspondingly, Swanston et al. [29] proposed that, by restricting starch synthesis, the genetic loci for waxy and high amylose traits may have similar, additive effects. In this respect, it is worth noticing that starch is composed of a mixture of two polysaccharides: amylose and amylopectin. Normal starch contains about 75% amylopectin and 25% amylose [1,30]. High-amylose barley has ≥40% amylose in its starch [1]. Oppositely, waxy starch has a low level of amylose (<5%).

Thus, barleys with increased β-glucan content were obtained by pyramiding favorable alleles at different loci underpinning starch alterations [29]. Even more striking, a recombination of the high-amylose allele (*amo1*) from the barley mutant Glacier AC38 and the *waxy* allele of waxy mutant Yon-M-kei 286 produced lines with noticeably higher β-glucan content (12.4%) than its parents carrying a single mutation [31]. These genotypes have found interest in the specialized waxy barley market [32]. Another high β-glucan barley variety, Beta-One, was developed by crossing Glacier AC38 with Shikoku Hadaka 97, which derives from Yon-M-kei 286, and was reported to contain 11.4% β-glucan [33]. In a subsequent extensive trial, however, the β-glucan content of Beta-One was found to be 8.4% [16].

Although the chemical composition of barley greatly varies among genotypes and in consequence of environmental conditions, the whole grain of common cultivated barley varieties typically contains 53–67% starch, 14–25% dietary fiber (which includes 3–7% β-glucan), 9–17% protein, 2–3% lipid, 1.4–2.5% ash, 0.8–2.2% low-molecular-weight sugars, 0.26–0.52% phosphorous, 0.03–0.06% calcium and 0.01–0.08% sodium [1,2,25,34,35].

In this work, we performed Quantitative Trait Loci (QTL) mapping analysis for the content in β-glucan and other barley grain components using a recombinant inbred line (RIL) population derived from a cross between Glacier AC38 and CDC Fibar, where the latter is a waxy cultivar with high β-glucan content. QTL analysis is a statistical method that identifies associations between phenotypic traits and specific genomic regions (loci; typically pinpointed by molecular markers) in order to explain the genetic basis of phenotype variation. QTL mapping of biparental segregating populations is a powerful tool for unraveling genes underpinning complex traits; thus, it plays an important role in gene cloning and characterization [36]. The objective of this study was, therefore, to identify loci that underpin the diversity in grain composition—particularly in β-glucan content—observed between the parents and, then, among the progeny lines of the above-mentioned cross. The research includes (i) phenotyping for traits under study; (ii) development of a high-density genetic map; (iii) identification of QTL regions; (iv) a comparison of these QTL with related known genes; and (v) the identification of candidate genes within the physical positions of the stable QTL.

## 2. Results and Discussion

### 2.1. Phenotyping

The two parents and 181 F7 RILs of the cross Glacier AC38 × CDC Fibar were analyzed for the content in β-glucan and other components of the grain (Table 1). As the covered/hulless trait has an obvious effect on grain composition [1,25], the averages for the genotypes grouped according to the two types of grain are also shown in Table 1, and the effect of this trait was assessed.

The barley RIL population under study showed relevant variability for all the characters (Table 1). The covered/hulless trait had a highly significant (*p* ≤ 0.001) effect on all traits but protein. Indeed, the hull accounts for approximately 13% of the mature grain dry weight [37] and its absence causes a proportional increase in the other grain components. Hulless barleys often have higher β-glucan contents than covered feeding barley, and the former are mainly used as human food because of easier edibility [16,38].

Although hulless genotypes had higher levels of phosphorous, calcium and sodium than those of covered genotypes, their ash content was lower (Table 1). The higher ash content in covered varieties is mainly due to the hull [25], which is comprised of about 6.0% ash, which, in turn, contains almost all the silicon present in the barley grain [1]. Higher phosphorus content in the grain of hulless barleys vs. covered ones was also observed by Bleidere and Grunte [35]. Minerals—mainly potassium, phosphorus and calcium—mostly accumulate as phytates in globoids, spherical solid inclusions embedded in protein bodies, predominantly located in the aleurone layer of the grain [39]. It is not clear, however, if differences in mineral contents between hulless and covered lines (Table 2) were associated with greater accumulation in the aleurone layer.

The genetic and environment effects (over two growing years) were evaluated with analysis of variance (ANOVA, Table 2).

The coefficient of determination (R^2^) measures the proportion of variation observed for each grain component that is explained by the statistical model considering genotype and growing year as factors determining the components. As R^2^ is a number between 0 and 1 with higher values corresponding to better goodness of fit, this statistic indicates that, apart from protein, variation in all grain component contents was satisfactorily explained by the two factors used in the statistical model, with the model fit varying from being moderately satisfactory for starch to very good for β-glucan. A good model fit (that is, high R^2^) is a favorable proviso for a successful QTL analysis.

The Coefficient of Variation (CV) is calculated from ANOVA as 100 times root MSE divided by the mean percentage of the grain component. It is, therefore, a measure of the unexplained variability normalized to the mean of each character. For example, even though β-glucan content had a larger R^2^ (0.97) than starch content (0.75), it also had a larger unexplained variation (i.e., CV was 9.5% and 4.4%, respectively) because the range of values (Table 1) for β-glucan (4.3–18.4%) was relatively much broader than for starch (51.4–65.2%) with respect to their means, though the widths of their ranges of variation were similar. Anyway, even the unexplained variability in β-glucan content was modest (CV < 10%).

Table 2 also gives the significance probabilities for the effects of the genotype and growing year. Genotype had a highly significant effect (*p* ≤ 0.001) for all traits but protein. Assuming *p* = 0.05 as the significance threshold, the significance of the genotype effect for protein was just beyond it, suggesting that a genetic effect probably existed, but it was quite weak in the studied population.

The year effect too was highly significant (*p* ≤ 0.001) for all traits but β-glucan, starch and protein (Table 2). It was, however, very significant (*p* ≤ 0.01) for protein and significant (*p* ≤ 0.05) for starch. It was not significant for β-glucan content (*p* = 0.30), meaning that, at least under our conditions, β-glucan content was mostly determined by genotype and not by the environmental difference over the two years of testing. As β-glucan content also had the highest R^2^, we can safely assume that its determination was mainly genotypic, which ensures stability of QTL effects. Breeding for high β-glucan barley, indeed, is a valuable and effective task [2,32,40].

As regards broad-sense heritability (H^2^), the highest value (0.97) was achieved for β-glucan content (Table 2), confirming its genotypic determination as already hinted by the ANOVA parameters. Protein content showed the lowest value (0.20), substantiating its low genetic effect in the present study. The other grain traits ranged across these two extreme values.

Correlation analysis showed some interesting relationships between grain traits (Appendix A). Calcium, sodium and phosphorous contents strictly correlated as they are co-components of phytate [39]. A noticeable positive correlation (r = 0.69, *p* < 0.0001) between β-glucan and lipid contents was observed (Figure 1). A positive association between β-glucan and lipid contents had been found in oat too [41].

A high β-glucan content (10.3%) was confirmed for CDC Fibar (Figure 2), closely in between the values found by Tonooka [32] and Izydorczyk et al. [42], and within the range of values reported by Rossnagel et al. [43] for five years of testing. The high-amylose Glacier AC38 barley was also relatively high in β-glucan in our study (7.4%), a value near to that found by Fujita et al. [31] and Oscarsson et al. [25], namely 7.9% in both cases. It also appears that the progeny of the cross Glacier AC38 × CDC Fibar showed transgressive segregation for β-glucan content; that is, they showed genetic variation well beyond that occurring between their parents (Figure 2). Thus, combining these genomes had an effect even more remarkable than that found by Fujita et al. [31].

### 2.2. Construction of the Glacier AC38 × CDC Fibar Genetic Linkage Map

Out of 40,246 high-quality SNP markers, 12,728 SNPs were polymorphic between the parents. After elimination of unlinked loci, the genotype data relating to 12,463 informative SNPs were assembled into eight linkage groups corresponding to the seven barley chromosomes (Table 3; Appendix A). More than one linkage group was obtained for chromosome 3H. The overall length of the map was 2526.53 cM with individual chromosome genetic length ranging from 256.86 cM (chromosome 3H) to 515.09 cM (chromosome 2H) and an average of 360.93 cM. The total number of mapped loci per chromosome ranged from 1379 (chromosome 6H) to 2302 (chromosome 5H) with an average of 1780.43 loci per chromosome (Table 3; Appendix A). Details about the genetic distances between markers are provided in Appendix A. The genome-wide mean inter-locus separation was 0.20 cM, varying from 0.17 cM (chromosomes 3H) to 0.24 cM (chromosome 2H). Furthermore, if we do not consider the co-segregant markers, the density ranged from 0.75 to 1 (chromosomes 1H and 7H respectively; Table 3; Appendix A).

A total of 45 QTL were mapped: thirty QTL out of 45 were identified based on BLUPs from single-environment data and 15 QTL based on BLUPs from multi-environment analysis (Appendix A). The highest number of QTL was identified for starch (nine), followed by ash (seven), lipid and β-glucan (six). Considering all the QTL identified, the proportion of the phenotypic variation explained ranged from 4.25% (*QSt_MEnv_5H*) to 74.62% (*QSo_2022_7H*). Among all these QTL, it is interesting to note the presence of a new QTL for β-glucan content on chromosome 7H.

For all traits except protein, nine associated regions were found in all environments tested (years 2021 and 2022) and in the multi-environment (MEnv) analysis, suggesting that stable QTL were identified.

### 2.3. QTL Assessment

As recombination of additive alleles is the most common cause of transgressive segregation, we supposed that recombination of parental alleles resulted in new haplotypes for β-glucan content at two or more loci. QTL analysis, indeed, showed that two main QTL were involved: *QBg_MEnv_1H* and *QBg_MEnv_7H*, on chromosome 1H and 7H, respectively (Table 4). These two QTL explained a relevant portion of the observed variability in β-glucan content: 28% and 27%, respectively (Table 4). Assuming their effects are roughly additive and quantitatively similar (Table 4), the distribution of β-glucan content across the barley population (Figure 2) ought to represent a composite of the four haplotypes obtained as combinations of the two parental alleles at the two loci. These haplotypes should be approximately equal in number and appear as three sub-populations: a sub-population of genotypes (nearly ¼ of the whole population), with a peak of frequency (mode) at about 6% β-glucan content, that carry the alleles associated with low β-glucan content at both loci; roughly half (slightly more, actually) of the population having one allele for high β-glucan content at either locus, with an average β-glucan content approximately 8.3%; and a sub-population (less than ¼ of the whole population, apparently) carrying the alleles for high β-glucan content at both loci, and displaying a broad peak with the average at 14–16% β-glucan content. Interestingly, there seems to be a sharp drop of genotype frequencies around 12% β-glucan, suggesting that this percentage represents a threshold useful for the identification of high β-glucan genotypes.

In addition, the strong increment of β-glucan content from the sub-population with only one allele for high β-glucan content to the sub-population with two alleles for high β-glucan content, with respect to the increment of β-glucan content from the sub-population with no alleles for high β-glucan content to the sub-population with one allele for high β-glucan content, suggests a synergistic effect between the alleles for high β-glucan content at the two loci. Correspondingly, Swanston et al. [29] found that pyramiding the high-amylose *amo1* allele and the *waxy* allele of waxy mutant Waxy Hector had additive effects on a range of quality characters but the effect on β-glucan content exceeded a simple additive effect.

Peak markers *BOPA2_12_30438* and *JHI-Hv50k-2016-452747* were used to identify the alleles of the QTL for β-glucan content at chromosomes 1H and 7H, respectively. The empirical remarks made on the full set of phenotype data (Figure 2; *N* = 183) were, then, confirmed by comparing the four haplotypes obtained as combinations of the alleles of CDC Fibar and Glacier AC38 at the two loci identified on chromosomes 1H and 7H (Figure 3; *N* = 163, only RILs for which the alleles at both markers were resolved have been included). In the progeny, the haplotypes corresponding to the two parents—AA (CDC Fibar) and BB (Glacier AC38)—were not significantly different (*p* = 0.0761; which, anyway, suggests that a larger experiment might show a, albeit small, real difference), whereas all the other pairwise comparisons showed very highly significant (*p* < 0.0001) differences for β-glucan content. Considering that the β-glucan content distributions of AA and BB haplotypes largely overlap, so that they are not phenotypically distinguishable, the haplotype medians showed in Figure 3 collimate quite well with the peaks of frequency (modes) observed in Figure 2.

In the parents, which bear only one allele associated with higher β-glucan content across the two QTL, the allele of CDC Fibar at locus *QBg_MEnv_7H*, which is associated with higher β-glucan content, would actually seem to be more effective than the Glacier AC38 allele for higher β-glucan content at locus *QBg_MEnv_1H* (Figure 2), but this ought to depend on their different genetic backgrounds, as a single peak at about 8.3% β-glucan content, rather than two separate peaks—one for each of the two sub-populations with only one allele for higher β-glucan content across the two loci—was observed (Figure 2). As mentioned, anyway, the difference in β-glucan content between the parental haplotypes, though overall small in the RILs, is close to being significant. There is, indeed, no causal reason for which they should have the same effect.

Additionally, minor effects of other genes (not identified by the QTL analysis) may, indeed, explain the amplitude of the broad peak corresponding to the sub-population with both alleles for high β-glucan content at the two major loci identified here. More exactly, this latter sub-population seems to have a width similar to that of the sub-population with only one allele for higher β-glucan content across the two loci, but—being numerically smaller—it reveals a platykurtic distribution, which suggests the existence of additional fixed effects (i.e., minor genes) over random ones. For example, as seen, the waxy mutation ought to have an effect by itself.

In this respect, it is worth noting that QTL *QBg_MEnv_7H* is close to gene *Wx-1* (*Horvu_PLANET_7H01G061200*), which is located at 17 Mbp on chromosome 7H, just outside the interval identified for *QBg_MEnv_7H* (Figure 4; Appendix A). *Wx-1* encodes Granule-Bound Starch Synthase I (GBSSI), the enzyme responsible for the synthesis of amylose, and whose mutations are responsible for the switch from normal to low/null amylose content [44]. As many Canadian barleys with high β-glucan content are also waxy, and the *QBg_MEnv_7H* allele causing higher β-glucan content is from the Canadian parent CDC Fibar, it may be speculated that the genetic association between *QBg_MEnv_7H* and *Wx-1* could have favored the selection of waxy barleys with particularly high β-glucan content even though their mutations at the *Wx-1* locus do not severely restrict starch accumulation. Not all waxy mutations, indeed, cause an increase of β-glucan content [17,26].

Although our main target was to study β-glucan content, so that we focus the discussion of our findings chiefly on the two QTL associated with this trait, we found QTL for all the studied grain traits (Table 4). Overlapping QTL explaining a large percentage of variability (R^2^; Table 4) observed for starch, lipid, ash, phosphorous, calcium and sodium were found on chromosome 7H, with a peak at 539–543 Mbp (Figure 4). As gene *Nud* (*Horvu_PLANET_7H01G578300*), responsible for the covered/hulless trait, is located at 542 Mbp on this chromosome, it was assumed as the prominent candidate to explain the common effects of these QTL. Some of these effects, are, indeed, known to be associated with the presence/absence of the hull [1,25,35]. Hulless genotypes also have a higher β-glucan content than covered ones (Table 1) [1,25]. However, like in the study of Steele et al. [40], no significant association of β-glucan content with the gene for covered/hulless grain (*Nud*) was observed. This is probably because the variability of β-glucan content was much higher than the effect of the hull, with large genetic variation within each of the two barley groups.

Minor QTL were also found for starch on chromosome 5H, protein on chromosome 2H, protein and ash on chromosome 4H (Table 4).

### 2.4. Candidate Genes for β-Glucan Content

As for β-glucan content, *QBg_MEnv_1H* encompasses a wide region of >200 Mbp with a marker peak at about 95 Mbp (Figure 4). Since gene *HvCslF9* (*Horvu_PLANET_1H01G145500*) is apparently involved in the synthesis of β-glucan [8,23] and is located approximately at 96 Mbp on the short arm of chromosome 1H [19], it was individuated as a good candidate gene for this QTL. The large extension of *QBg_MEnv_1H* is most probably due to the *HvCslF9* gene mapping near the centromere of chromosome 1H [20], since recombination is low around the centromere [45,46]. In detail, the region identified by *QBg_MEnv_1H* spanned from 84 to 292.8 Mbp and, considering that the 1H centromere on Morex v3 is located at about 206 Mbp [47], with high probability this QTL overlaps with the pericentromeric region.

Various QTL and GWAS studies [20,21,48,49] identified a region on chromosome 1H involved in β-glucan content that encompasses *HvCslF9*. Specifically, in this respect, a genome-wide association study (GWAS) on grain β-glucan content [21] confirmed that a QTL identified previously by Han et al. [48] for malt β-glucan content on chromosome 1H co-locates with *HvCslF9*. As *HvCslF9* is expressed during early grain development [20], it might contribute to β-glucan synthesis in a temporally and/or spatially restricted manner in some genotypes [23]. Thus, Garcia-Gimenez et al. [23] concluded that *HvCslF9* is unlikely to influence mature grain β-glucan content in the Golden Promise cultivar, but it might contribute to β-glucan biosynthesis in other genotypes.

The main gene for grain β-glucan content, *HvCslF6*, is located on the long arm of chromosome 7H near the centromere [19]. However, we did not find a QTL in this region; rather, *QBg_MEnv_7H* was detected on the short arm of chromosome 7H (Figure 4), over a narrow interval (4 Mbp) with marker peak at approximately 20 Mbp (Figure 4). As no loci involved in β-glucan synthesis are known in this genome region, we highlighted gene *Horvu_PLANET_7H01G069300*, a gene for an ATP-Binding Cassette (ABC) transporter located close to the QTL peak (Figure 4), as a suitable candidate for this QTL. *Horvu_PLANET_7H01G069300* is annotated as homologous to the rice gene *OsABCG51*, which codes for an ABC transporter G family member [50].

Although the choice of *HvCslF9* as a candidate for *QBg_MEnv_1H* seems obvious, *Horvu_PLANET_7H01G069300* was identified as a suitable candidate for QTL *QBg_MEnv_7H* because ABC transporters are ATPase-coupled transmembrane transporters that can move out, or in, the cell many kinds of molecules, including polysaccharides [51,52]. Specifically, according to the Transporter Classification Database (https://www.tcdb.org/ accessed on 3 June 2024), ABC transporters of the β-Glucan Exporter (3.A.1.108-GlucanE) family are efflux carriers that bacteria use to extrude β-(1→2)-glucans (https://enzyme.expasy.org/EC/7.5.2.3 accessed on 3 June 2024; [53]). Although transporters of this kind have not yet been found in higher plants, it can be expected that similar carriers are also present in these latter. This is because, whereas cellulose is assembled at the plasma membrane, matrix-phase polysaccharides, such as β-glucan, are usually synthesized in the Golgi apparatus and must be carried through the plasmalemma to reach the cell wall [6], although it cannot be ruled out that β-glucan synthesis occurs at the plasma membrane [8]. Thus, we speculated that a gene for β-glucan synthesis and an efflux carrier for β-glucan might explain the transgressive segregation for β-glucan content observed in the cross Glacier AC38 × CDC Fibar, with the allele associated with higher β-glucan synthesis provided by Glacier AC38 at *QBg_MEnv_1H* and the allele associated with higher β-glucan transport provided by CDC Fibar at *QBg_MEnv_7H*. Physiological cooperation between β-glucan synthesis and transport could also explain the above-mentioned synergistic effect between the alleles for high β-glucan content at the two loci identified in this study.

### 2.5. Other QTL Effects and Final Considerations

The QTL for starch, lipid and ash contents at the short arm of chromosome 1H largely overlapped (Figure 4), suggesting the haplotype at *HvCslF9* could have pleiotropic effects. If so, it could be noted that, while the effect on lipid content was in the same direction as the QTL for β-glucan content present in this cluster, *QBg_MEnv_1H*, QTL for starch and ash contents showed effects that were directionally opposite to the effect of this QTL for β-glucan content (Table 4). The former, positive association is linked to the observed positive correlation between β-glucan and lipid contents (Figure 1). The latter, negative association is presumably due to the inverse relationship typically observed between starch and β-glucan content [8], and, as regards ash, it might be consequent to the low content of minerals in β-glucan and lipid.

Thus, the overlapping QTL *QBg_MEnv_1H* and *QSt_MEnv_1H* on the short arm of chromosomes 1H have opposite effects on β-glucan and starch contents, respectively (Table 4). It seems obvious that, due to partitioning of photosynthates, starch synthesis competes with β-glucan synthesis for glucose [8,22]. Indeed, competing biosynthetic pathways explain why perturbations in starch metabolism positively affect grain β-glucan content [8]. Nevertheless, based on the present findings (Table 1), it appears that high β-glucan content (>12%) can be obtained even if the starch content is not strongly abated, since the negative correlation between starch and β-glucan contents was low in our genetic materials (r = −0.29; Appendix A). For example, the highest β-glucan content in the progeny was 18.4% (Table 1), but this same line had a starch content of 56.5% (Appendix A), which is not particularly low.

It should be noted, however, that gene *Horvu_PLANET_1H01G135400*, encoding for Starch Synthase 3a (SSIIIA), is located at about 85 Mbp on chromosome 1H, within the interval of QTL *QBg_MEnv_1H*. This is particularly relevant because the locus underlying the mutant Glacier AC38, named *Amo1*, was shown to be located on chromosome 1H [54] in the same region as the *SSIIIA* gene, which was proposed to be the involved candidate gene [55]. As evidence of this, sequencing of the *SSIIIa* gene showed an SNP causing an amino acid substitution in the *amo1* mutant (i.e., Glacier AC38) compared with the original Glacier cultivar [55]. As the mutation causing the high-amylose content of Glacier AC38 has been inferred to be due to a single gene [56], and the original Glacier cultivar has a lower β-glucan content than the mutant [25], it is possible that the effect of *QBg_MEnv_1H* is due to the mutated *SSIIIA* gene rather than to a specific allele at *HvCslF9*, even though the latter is much closer to the peak of QTL *QBg_MEnv_1H* (Figure 4). Unless, of course, two closely associated mutations have accumulated in Glacier AC38. An alternative hypothesis is that the allele for high β-glucan content contributed by Glacier AC38 in this study was already present in the original cultivar Glacier and, therefore, it is different from the *amo1* locus. Its effect would only now be noticed because of its unique interaction with the allele at locus *QBg_MEnv_7H* provided by CDC Fibar. It is, indeed, very possible that both genes affect β-glucan content, and, being in linkage, the minor effect of the *amo1* locus is indistinguishable from the main effect of *HvCslF9* in our materials.

Although we could not precisely identify the genes involved in the increase of β-glucan content following the transgressive recombination observed in the studied cross, we found two QTL that can be of great interest in breeding programs for high β-glucan barley. Further studies are, however, necessary to elucidate the several still unknown aspects of the genetic determination of barley β-glucan content.

## 3. Materials and Methods

### 3.1. Barley Materials

CDC Fibar is a two-row, hulless, Spring waxy barley cultivar developed at the Crop Development Centre, Saskatoon, Saskatchewan (CAN) as a specialty food barley with high β-glucan content and plump grains [32,43,57]. It carries a single nucleotide change at the gene (*Wx-1*) coding for Granule Bound Starch Synthase I (GBSSI) [58,59].

The Pentlandfield (UK) stock of the Canadian cultivar Glacier was found at the Scottish Plant Breeding station as a spontaneous mutation (designated AC38) of the six-rowed, covered, Spring cultivar Glacier that contains a strongly increased percentage of amylose in its starch [30]. The high amylose characteristic in Glacier is controlled at a single locus with an additive dose effect of alleles [56]. Even this mutation does not affect plumpness and test weight [60]. However, the total starch content of Glacier AC38 is reduced by some percent points with respect to the original Glacier cultivar [61], while β-glucan content is increased [25]. The size of starch granules is, however, smaller than in the original cultivar, with less regular shaping [56].

The offspring of the barley cross Glacier AC38 × CDC Fibar was bred up to the F5 plant generation in the field at CREA—Research Centre for Genomics and Bioinformatics in Fiorenzuola d’Arda (Piacenza, Northern Italy) with the pedigree method: first (F1–F2), by space planting in a single row, and then by sowing progeny lines (each tracing back from a single F2 plant) in 1 m rows. Thereafter, a single seed from each of the 220 F5 progeny lines plus the parents was grown into an (F6) plant in a greenhouse and genotyped for QTL mapping. These recombinant inbred lines (RILs) were grown in a greenhouse for two further years to produce enough grains for phenotyping. Specifically, seeds harvested in 2019 from the individual F6 plants were used, in both 2021 and 2022, to grow three to four plants of each progeny line for analyses of β-glucan and other grain components. Ultimately, the mapping population consisted of 181 RILs plus the parents, as some progeny lines did not produce enough seed for testing, and it was further reduced to 177 RILs as a few were filtered out because their genotyping was not satisfactory.

### 3.2. Phenotyping: Barley Grain Analyses

Barley seeds were ground to 0.5 mm with a Cyclotec Sample Mill (Foss Italia S.p.A., Padova, Italy). β-Glucan content was determined by means of enzymatic analysis in 2021, and with Near-InfraRed Spectroscopy (NIRS) in 2022, as detailed below.

Protein, starch, lipid, ash, phosphorus, calcium and sodium were analyzed with NIRS in both 2021 and 2022.

#### 3.2.1. Enzymatic Analysis of β-Glucan

In 2021, β-Glucan content was assessed by means of the mixed-linkage β-glucan assay kit (K-BGLU; Megazyme, Bray, Ireland) according to the streamlined procedure of McCleary and Codd [62]. This method has been adopted by AOAC International (Method 995.16), AACC (Method 32-23.01) and ICC (Method No. 166). Absorbance measurements were performed at a wavelength of 510 nm using a DU-730 spectrophotometer (Beckman Coulter Inc., Brea, CA, USA). Moisture was determined using an HA60 IR thermobalance (Precisa Instruments, Diekinton, Germany). β-Glucan content was expressed as percentage of flour on a dry weight basis (dwb).

#### 3.2.2. NIRS Analysis

In 2022, NIR log 1/R (pseudo-absorbance) spectra were obtained from each sample with an NIR DS2500 F spectrometer (FOSS, Hillerød, Denmark) with a spectral range of 850–2500 nm, by using approximately 3 g of flour placed in a FOSS mini open ring cup cell. Each spectrum is the average of five spatial subsamples analyzed during automatic ring-cup rotation, each obtained from 32 sub-scans by default. The spectral signal was automatically translated into an analytical value by using a predictive model for β-glucan developed from the analytical data obtained in 2021 [63] and predictive models provided by FOSS Italia srl (Padova, Italy) for the other grain traits. The latter were based on the following laboratory reference procedures: Kjeldahl, polarimetric, Soxhlet and muffle furnace, for protein, starch, lipid and ash, respectively. Predictive models for phosphorus, calcium and sodium had been developed with Atomic Absorption Spectroscopy (Personal communication [64]). FOSS ISIscan Nova software package (version 8.10.2.12) was used for the operation of the NIR spectrometer and data acquisition. Analyses were performed in duplicate for each milled sample.

### 3.3. DNA Extraction

One developed leaf was sampled from the parents and each F6 RIL and ground using the Retsch MM300 Mixer Mill instrument (Newtown, PA, USA). Then, genomic DNA was extracted and purified using a DNeasy Plant Mini Kit (Qiagen, Milan, Italy) following the manufacturer’s instructions. Evaluation of the quality and quantity of the extracted DNA was performed using a Qubit™ fluorometer with the Qubit™ dsDNA BR Assay kit (Invitrogen, by Thermo Fisher Scientific, Monza, Italy).

### 3.4. Linkage Analysis

The single-nucleotide polymorphism (SNP) molecular markers were used to analyze the parents and the RILs. Genotyping was performed by TraitGenetics (SDS-TraitGenetics GmbH, Gatersleben, Germany) with the Infinium iSelect 50K barley SNP BeadChip array (Illumina Inc., San Diego, CA, USA) carrying 44,040 functional markers [65]. Samples were filtered and four RILs were removed from the dataset used for linkage analysis because they had >30% missing genotypic data. SNP markers with ambiguous SNP calling between parents and/or with a negative hybridization response in most lines were removed from the data set; after this check, 40,246 high-quality SNP markers were retained. Linkage analysis was performed using R/ASMap (version 1.0-6) [66,67] with an LOD score threshold of 9.0, maximum distance of 10 cM and the Kosambi mapping function to calculate map distances [68]. The obtained linkage groups were assigned to chromosomes using BLAST search matches of the corresponding SNP sequences on two barley reference genomes (cv Morex v3 and RGT Planet v1 [47]). Within each linkage group, the best order of markers and the genetic distances were found setting the *p* value to 1 × 10^−9^.

### 3.5. QTL Analysis

For QTL analysis, best linear unbiased predictions (BLUPs) for each single environment and across environments were calculated using the Restricted/Estimated Maximum Likelihood method implemented in R/ASReml [69]. All the predictor variables were fitted as random terms of the linear mixed model and the obtained BLUPs were used to perform QTL mapping.

QTL mapping was conducted with the R/qtl package of the R statistical computing software (version 1.60) [70] using the procedure described in Desiderio et al. [71]. For each trait, an initial QTL scan was performed using simple interval mapping with a 1 cM step [72], and the position of the highest LOD was recorded. A genome-wide significance level of 5% was calculated after 1000 permutations [73], and an LOD threshold greater than or equal to 3.2 was used to declare a QTL. Then, the position and the effect of the QTL were determined using the multiple imputation method by executing “sim.geno”, “fitqtl”, “addqtl” and “refineqtl” commands [36]. QTL interactions were studied, and, if the significant locus combinations were identified, they are reported based on F-measure. The additive effects of QTL were estimated as half of the difference between the phenotypic values of the respective homozygotes. The confidence interval (CI) of each QTL was determined as proposed by Darvasi and Soller [74]. Next, for each trait, QTL found using data from single environments and multi-environment data were considered to correspond to the same QTL if CIs were overlapping and the additive effect was conferred by the same parent. QTL were named according to the rule “*Q* + trait code + environment + chromosome”, where *Q* stands for QTL, ‘trait code’ refers to the trait acronym presented in Table 1, ‘environment’ stands for ‘2021’, ‘2022’ and ‘multi-environment’ (MEnv) and ‘chromosome’ is the barley chromosome on which the QTL is located.

### 3.6. Analysis of Physical Regions Carrying QTL

To analyze the QTL identified in this work in detail, we decided to project the high-density genetic map onto the Morex v3 reference genome. For this purpose, all the sequences of the markers were located on the genome of Morex v3 using BLAST search matches. The alignment of genetic and physical positions revealed a consistent inverted region on the long arm of chromosome 7H. Based on this finding, and considering that this inversion, on chromosome 7H, is known, and well characterized, only among Morex3 and RGT Planet, we decided to use RGT Planet as the barley reference genome in this work [47]. The physical intervals of all QTL regions, defined by peak markers and flanking markers corresponding to the confidence intervals (CIs), were determined in accordance. The physical interval retrieved for all significant regions was then used to compare the QTL identified in this work with known genes/QTL, previously identified in barley [19,20,21,22,48,49]. For this purpose, the physical interval of known genes was defined using their sequence as a query to perform the BLAST searches against the RGT Planet reference genome. Finally, the physical regions underlying the QTL were inspected to identify candidate genes based on their functional annotation.

### 3.7. Statistical Analysis

Basic statistical analysis was generated using SAS^®^ software (version 9.04.01M7P08062020; SAS^®^ Studio release 3.81—SAS OnDemand for Academics; copyright © 2012-2020, SAS Institute Inc., Cary, NC, USA). Analysis of Variance (ANOVA) was performed with the General Linear Model (GLM) procedure according to (1) a one-way ANOVA model based on genotype values averaged over the two years, to assess the effect of covered/hulless grain; (2) a two-way ANOVA model without interaction using genotype and year as fixed and random factors, respectively. Although data were replicated within each year, these were technical duplicates, not field replications. Using them to calculate the Mean Square error variance (MS_e_) would have, therefore, caused an under-estimation of the actual field MS_e_ and would have, thereby, inflated estimations of *p* values. By considering genotype a fixed effect and including the interaction effect into MS_e_, more cautious and reliable inferences can be drawn if true biological replications are not available [75].

Broad-sense heritability (H^2^) was calculated for each trait over the two years of the experiment from the variance component estimations [76] obtained in the latter ANOVA model, according to the following formula:H^2^ = σ^2^_g_/(σ^2^_g_ + σ^2^_ε_/n_y_),(1)
where σ^2^_g_ represents the genotypic variance and σ^2^_ε_ is the error term; n_y_ is the number of years (two). The formula does not explicitly consider σ^2^_gy_—the variance component of the genotype by environment (year) interaction—because σ^2^_gy_ was conflated into σ^2^_ε_. This calculation also does not incorporate σ^2^_y_ (the variance component for the year) because the classical calculation of H^2^ focuses on the ranking of genotypes rather than on absolute performances, and, therefore, it does not include purely environmental variance components [77]. The traditional H^2^, indeed, corresponds to the restricted broad-sense heritability as defined by Gordon et al. [78]

The Pearson’s correlation coefficient (r), which measures the strength and direction of linear relationships between pairs of continuous variables, was estimated using the CORR procedure of SAS^®^ software (version 9.04.01M7P08062020).

For the comparison of β-glucan levels among haplotypes, Kruskal–Wallis non-parametric one-way ANOVA was performed by using the NPAR1WAY procedure of SAS^®^ software (version 9.04.01M7P08062020) (since the Brown–Forsythe test of the GLM procedure showed significant heteroskedasticity among haplotypes). Statistical significance of differences among haplotypes was evaluated with the Dwass, Steel, Critchlow-Fligner (DSCF) post-hoc multiple comparison test.

## Figures and Tables

**Figure 1 ijms-25-06296-f001:**
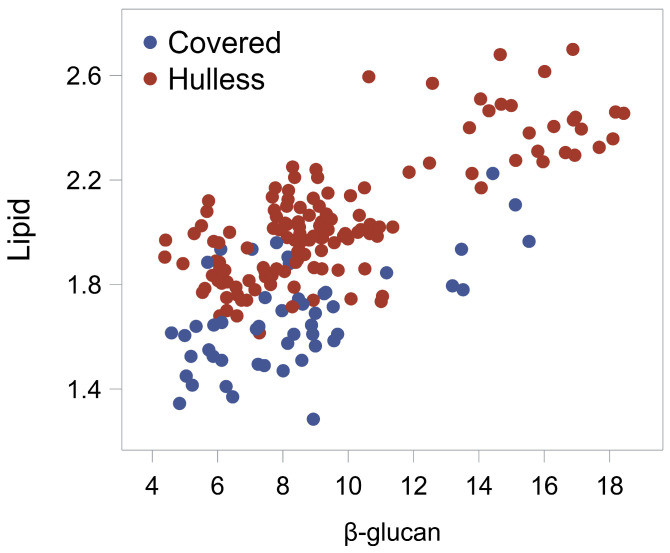
Scatter plot of lipid vs. β-glucan contents for the RIL population. Covered and hulless genotypes are distinguished by different colors of the dots.

**Figure 2 ijms-25-06296-f002:**
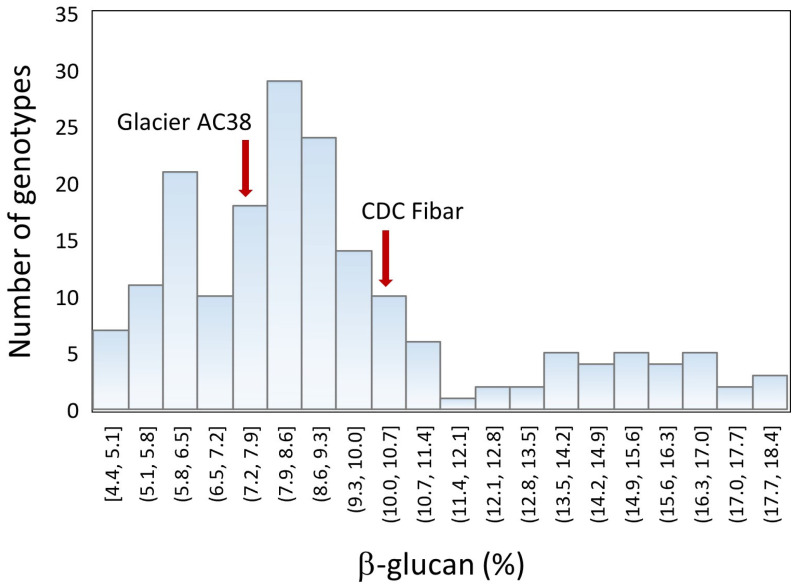
Distribution of genotypes according to their β-glucan content. The y-axis shows the number of genotypes whose β-glucan content falls in the range indicated on the x-axis for each histogram.

**Figure 3 ijms-25-06296-f003:**
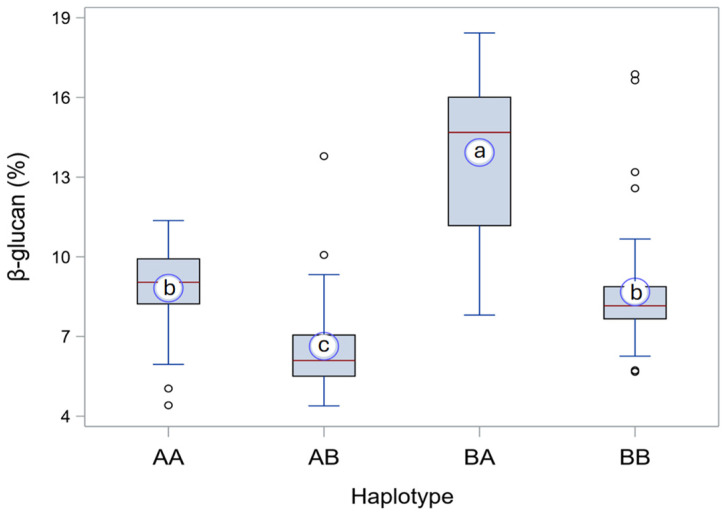
Box plot of β-glucan content for the four haplotypes with alleles of either CDC Fibar (allele ‘A’ in haplotype tags) or Glacier AC38 (allele ‘B’) at *QBg_MEnv_1H* (the former locus in the haplotype tag) and *QBg_MEnv_7H* (the latter locus in the tag) in the offspring of the barley cross Glacier AC38 × CDC Fibar. Each box indicates the interquartile range (IQR); that is, the range from the first (lower) quartile (Q1) to the third (upper) quartile (Q3) of the rank-ordered data of each haplotype (where the k^th^ quartile is the score below which k quarts of the data frequency distribution fall). The whiskers that extend from each box indicate the minimum and maximum observed values that are outside of the IQR but within a distance ≤ 1.5·IQR (a conventional threshold beyond which data are considered outliers) either below the lower (Q1) or above the upper (Q3) edge of the box, respectively. Outliers (circles) are observations that are more extreme than 1.5·IQR either below the Q1 edge or above the Q3 edge. The red line inside each box indicates the median value (or second quartile, Q2). The large blue circle with white filling in each box represents the mean value, and the lower-case letter inside it indicates significance of difference among haplotypes: haplotypes with the same letter are not significantly different in their β-glucan content (*p* ≤ 0.05; DSCF test). Numbers (*n*) of RILs in each haplotype: AA, *n* = 48; AB, *n* = 39; BA, *n* = 33; BB, *n* = 43.

**Figure 4 ijms-25-06296-f004:**
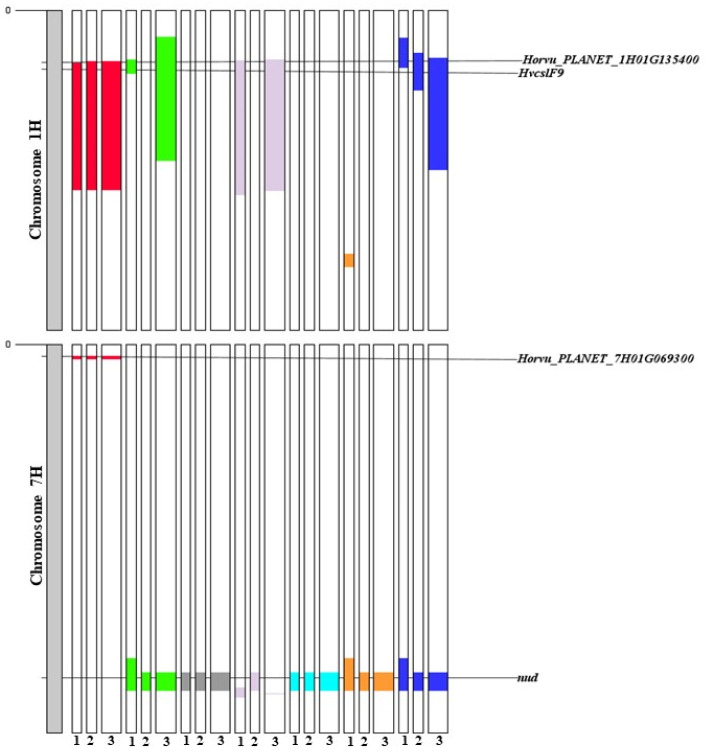
Distribution of co-localized QTL detected in this work on chromosomes 1H and 7H. Known genes co-localized with our QTL, and the putative candidate genes, have been reported. QTL obtained using all environments tested (2021, named as ‘1’; 2022 as ‘2’) and in the multi-environment (MEnv, ‘3’) are reported. The colors represent the different phenotypic traits: red stands for β-glucan, green for ash, grey for calcium, light purple for lipid, light blue for phosphorus, orange for sodium and blue for starch.

**Table 1 ijms-25-06296-t001:** Measured values of grain components for the mapping population (genotype values were averaged across the two years of the study).

Grain Component	Acronym	Mean (% dwb ^1^)	Minimum (% dwb)	Maximum (% dwb)	Average for Covered Genotypes (*N* = 49)	Average for Hulless Genotypes (*N* = 134)	Effect of Covered/ Hulless (*p*)
β-Glucan	Bg	9.3	4.3	18.4	8.2	9.6	<0.0001
Starch	St	58.3	51.4	65.2	54.7	59.6	<0.0001
Protein	Pr	11.5	7.2	18.6	11.4	11.5	0.8284
Lipid	Li	1.9	1.3	2.7	1.7	2.0	<0.0001
Ash	As	2.9	1.9	4.2	3.3	2.7	<0.0001
Phosphorus	Ph	0.496	0.235	0.750	0.326	0.559	<0.0001
Calcium	Ca	0.100	0.055	0.155	0.073	0.110	<0.0001
Sodium	So	0.013	0.005	0.020	0.009	0.015	<0.0001

^1^ Dry weight basis.

**Table 2 ijms-25-06296-t002:** Analysis of variance and broad-sense heritabilities (H^2^).

Grain Component	R^2^	CV (%)	Genotype Effect (*p*)	Year Effect (*p*)	H^2^
β-Glucan	0.97	9.5	<0.0001	0.2952	0.97
Starch	0.75	4.4	<0.0001	0.0205	0.67
Protein	0.56	21.1	0.0640	0.0031	0.20
Lipid	0.85	9.3	<0.0001	<0.0001	0.79
Ash	0.85	14.5	<0.0001	<0.0001	0.62
Phosphorus	0.92	17.2	<0.0001	<0.0001	0.76
Calcium	0.92	14.9	<0.0001	<0.0001	0.73
Sodium	0.81	23.8	<0.0001	<0.0001	0.48

**Table 3 ijms-25-06296-t003:** Distribution of molecular markers in the chromosomes of the Glacier AC38 × CDC Fibar barley map. Information on Kosambi centiMorgan (cM) length of the maps across the barley chromosomes, and chromosome-based mean inter-locus separation (cM/marker) is also provided.

Chromosome Linkage Group	cM	Total Markers	Total Markers (No Co-Segregant)	cM/ Marker ^1^	cM/ Marker ^2^
1H_1	349.54	1701	469	0.21	0.75
2H_1	515.09	2189	624	0.24	0.83
3H_1	244.6	1367	290	0.18	0.84
3H_2	12.26	166	21	0.07	0.58
3H_all	256.86	1533	311	0.17	0.83
4H_1	316.64	1630	336	0.19	0.94
5H_1	445.79	2302	528	0.19	0.84
6H_1	296.74	1379	330	0.22	0.90
7H_1	345.87	1729	345	0.20	1.00
All	2526.53	12,463	2943	0.20	0.86

^1^ Calculated considering all the markers. ^2^ Calculated considering only the non-co-segregating markers.

**Table 4 ijms-25-06296-t004:** QTL established across the two years of the experiment. Additive effects are referred to the Glacier AC38 vs. CDC Fibar comparison. Percentages are referred to the whole grain (dwb).

Trait	Chrom.	QTL Name	Peak (cM)	C.I. (cM)	LOD	R^2^	Additive Effect (%)
β-Glucan	1H	*QBg_MEnv_1H*	119.8	118.2–121.4	17.2	27.9	1.744
β-Glucan	7H	*QBg_MEnv_7H*	0	0–1.7	16.9	27.4	−1.725
Starch	1H	*QSt_MEnv_1H*	119.8	115.4–124.2	8.8	10.4	−1.018
Starch	5H	*QSt_MEnv_5H*	136.7	125.9–147.5	3.9	4.3	0.651
Starch	7H	*QSt_MEnv_7H*	191.4	190.4–192.4	30.3	48.3	−2.474
Protein	2H	*QPr_MEnv_2H*	125.2	120.9–129.5	4.9	10.7	−0.630
Protein	4H	*QPr_MEnv_4H*	232.5	227.7–237.3	4.4	9.5	−0.596
Lipid	1H	*QLi_MEnv_1H*	120.7	116.9–124.5	7.7	12.2	0.098
Lipid	7H	*QLi_MEnv_7H*	180.9	179.5–182.3	17.6	31.8	−0.174
Ash	1H	*QAs_MEnv_1H*	116.5	108.0–125.0	3.9	5.4	−0.112
Ash	4H	*QAs_MEnv_4H*	232.5	225.0–240.0	4.4	6.1	−0.116
Ash	7H	*QAs_MEnv_7H*	191.4	190.1–192.7	20.3	35.4	0.317
Phosphorous	7H	*QPh_MEnv_7H*	191.4	190.7–192.1	47.8	67.1	−0.113
Calcium	7H	*QCa_MEnv_7H*	191.4	190.6–192.2	36.6	61.4	−0.018
Sodium	7H	*QSo_MEnv_7H*	191.4	190.6–192.2	36.3	61.1	−0.003

## Data Availability

Data is contained within the article and Appendix A.

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
