# Peer review of "QTL Analysis of β-Glucan Content and Other Grain Traits in a Recombinant Population of Spring Barley"

_ijms, 2024, doi:10.3390/ijms25126296_

Round 1

Reviewer 1 Report

Comments and Suggestions for Authors

The authors of the manuscript ‘QTL Analysis of β-Glucan Content and Other Grain Traits in a Recombinant Population of Spring Barley’ developed a high-density SNP linkage map in spring barley recombinant inbred line populations and detected 45 QTLS for eight traits such as β-glucan, starch, protein, lipid, ash, phosphorous, calcium, and sodium. They also identified candidate genes involved in n β-glucan biosynthesis within the physical positions of the stable QTLs. The results are interesting, and the manuscript is well-written.

Specific comments:

Introduction: Please include information about the contents of the β-glucan, starch, protein, lipid, ash, phosphorous, calcium, and sodium among barley cultivars.

Please define the abbreviation when using it first.

L60: Poaceae (Non italic).

L60: cellulose synthase-like (Italic).

L214-219: Did the author identify novel QTL in this study? 

Comments on the Quality of English Language

 Minor editing of the English language required

Author Response

The authors of the manuscript ‘QTL Analysis of β-Glucan Content and Other Grain Traits in a Recombinant Population of Spring Barley’ developed a high-density SNP linkage map in spring barley recombinant inbred line populations and detected 45 QTLS for eight traits such as β-glucan, starch, protein, lipid, ash, phosphorous, calcium, and sodium. They also identified candidate genes involved in n β-glucan biosynthesis within the physical positions of the stable QTLs. The results are interesting, and the manuscript is well-written.

Thanks.

Specific comments:

Introduction: Please include information about the contents of the β-glucan, starch, protein, lipid, ash, phosphorous, calcium, and sodium among barley cultivars.

Done.

Please define the abbreviation when using it first.

Done.

L60: Poaceae (Non italic).

Done.

L60: cellulose synthase-like (Italic).

Done.

L214-219: Did the author identify novel QTL in this study?

We have clarified in the indicated paragraph that we found a novel QTL for β-glucan content.

We wish to thank again the Reviewer for the useful comments.

Reviewer 2 Report

Comments and Suggestions for Authors

The manuscript is quite interesting and falls under the general scope of the journal. The manuscript is correctly structured. The title clearly reflects the content. Some problems need to be fixed.

1.     Rewrite line # 30-31: Although barley (Hordeum vulgare L) is primarily used as feed…….

2.     You need to explain which physiological conditions make β-glucan more soluble than cellulose in line # 42-43.

3.     Rewrite the line # 66-67: In the starchy endosperm of…...

4.     Rewrite the line # 84-85 as the actual meaning is not clear:  When, instead, chemical mutagenesis was used to induce an……

5.     Delete the sentence in line # 90: At an earlier time of this same line of thinking,

6.     Provide brief details of GWAS analysis and its importance in introduction section.

7.     Write clear objective of the current study in the introduction section.

8.     Provide method in detail for the measurement of starch, phosphorus, lipid in section 3.2.

9.     Provide graphical representation of QTLs across chromosomes like MAPCHART tool in result section.

10.  Rewrite the line # 231: the barley population (Figure 2) ought to correspond to….

11.  Rephrase the line # 530-531.

12.  Compare you results with previously reported studies.

13.  Revisit and revise all the reference with uniform format. Follow a similar format in all the references.

14.  Line 189, what is the y-axis title for this figure? I feel the resolutions of the figures are not ideal, the authors may need to replace all the figures with a higher resolution.

Comments on the Quality of English Language

It won't hurt if the authors do another through check with English. 

Author Response

The manuscript is quite interesting and falls under the general scope of the journal. The manuscript is correctly structured. The title clearly reflects the content. Some problems need to be fixed.

Thanks for your careful revision.

  1. Rewrite line # 30-31: Although barley (Hordeum vulgare L) is primarily used as feed…….

Done.

  1. You need to explain which physiological conditions make β-glucan more soluble than cellulose in line # 42-43.

Done.

  1. Rewrite the line # 66-67: In the starchy endosperm of…...

Done.

  1. Rewrite the line # 84-85 as the actual meaning is not clear: When, instead, chemical mutagenesis was used to induce an……

We have re-written this sentence and moved it to the previous paragraph.

  1. Delete the sentence in line # 90: At an earlier time of this same line of thinking,

Done.

  1. Provide brief details of GWAS analysis and its importance in introduction section.

Aim and importance of QTL analysis have been briefly introduced.

  1. Write clear objective of the current study in the introduction section.

Done.

  1. Provide method in detail for the measurement of starch, phosphorus, lipid in section 3.2.

Done.

  1. Provide graphical representation of QTLs across chromosomes like MAPCHART tool in result section.

The graphical representation of QTLs across chromosomes was added as Supplementary Figure S1.

  1. Rewrite the line # 231: the barley population (Figure 2) ought to correspond to….

Done.

  1. Rephrase the line # 530-531.

Done.

  1. Compare you results with previously reported studies.

In the manuscript we did our best to quote the pertinent literature and discuss our findings in relation to the results of previous studies. Now, we have added five more references. If we have forgotten any relevant paper, we would be grateful to the Reviewer if she/he could point out our omissions.

  1. Revisit and revise all the reference with uniform format. Follow a similar format in all the references.

Done.

  1. Line 189, what is the y-axis title for this figure? I feel the resolutions of the figures are not ideal, the authors may need to replace all the figures with a higher resolution.

We have added the title of the y-axis in Figure 2. We strived to provide high-resolution figures as possible.

We wish to thank again the Reviewer for the useful suggestions.